# Taxonium, a web-based tool for exploring large phylogenetic trees

**Theo Sanderson***

The Francis Crick Institute, London, United Kingdom

**Abstract** The COVID-19 pandemic has resulted in a step change in the scale of sequencing data, with more genomes of SARS-CoV-2 having been sequenced than any other organism on earth. These sequences reveal key insights when represented as a phylogenetic tree, which captures the evolutionary history of the virus, and allows the identification of transmission events and the emergence of new variants. However, existing web-based tools for exploring phylogenies do not scale to the size of datasets now available for SARS-CoV-2. We have developed Taxonium, a new tool that uses WebGL to allow the exploration of trees with tens of millions of nodes in the browser for the first time. Taxonium links each node to associated metadata and supports mutation-annotated trees, which are able to capture all known genetic variation in a dataset. It can either be run entirely locally in the browser, from a server-based backend, or as a desktop application. We describe insights that analysing a tree of five million sequences can provide into SARS-CoV-2 evolution, and provide a tool at cov2tree.org for exploring a public tree of more than five million SARS-CoV-2 sequences. Taxonium can be applied to any tree, and is available at taxonium.org, with source code at github.com/theosanderson/taxonium.

## Editor's evaluation

Sanderson developed novel interactive software for visualizing phylogenetic trees representing millions of sequences. This is a fundamental advance over previous software that is typically limited to trees with a few thousand tips. Taxonium has been used intensively by the virus evolution community over the past months and has thus already proven its utility and performance.

**\*For correspondence:**
theo.sanderson@crick.ac.uk

**Competing interest:** The author declares that no competing interests exist.

## Introduction

Genomic researchers responded to the emergence of SARS-CoV-2 with rapid collaboration at unprecedented scale. Open protocols were quickly generated for amplicon sequencing (*Tyson et al., 2020*), and allowed researchers across the globe to produce ever-growing genomic datasets stored both in the INSDC databases (*Cochrane et al., 2016*), and in GISAID (*Shu and McCauley, 2017*), with the latter recently surpassing 11 million sequences. Importantly, new tools were also developed to understand the functional diversity of these samples, by assigning them into lineages proposed by the community (*Rambaut et al., 2020*; *O'Toole et al., 2021*), and allowing interactive exploration of trends in these data over time (*Hodcroft, 2021*; *Chen et al., 2021*; *Tsueng et al., 2022*).

The fundamental representation of a viral epidemic for genomic epidemiology is a phylogenetic tree, which approximates the transmission tree and can allow insights into the direction of migration of viral lineages (*Wohl et al., 2016*). When trees are annotated with the mutations that have occurred at each internal node, they capture information about parallel and convergent evolution, which can identify recurrent mutations of concern.

The unparalleled size of SARS-CoV-2 datasets has posed challenges for the array of tools that researchers have previously relied upon to manipulate, analyse, and visualise genomic data. In particular, the construction of phylogenetic trees, and the visualisation of these trees, have been major

**eLife digest** Since 2020, the SARS-CoV-2 virus has infected billions of people and spread to 185 countries. The virus spreads by making new copies of its genome inside human cells and exploits the cells' machinery to synthesise viral proteins it needs to infect further cells. Each time the virus copies its genetic material there's a chance that the replication process introduces an error to the genetic sequence. Over time, these mutations accumulate which can give rise to new variants with different properties. These new variants, originating from a common ancestor, may spread faster or be able to evade immune systems that have learnt to recognise previous variants.

To understand where new variants of SARS-CoV-2 come from and how related they are to each other, scientists build family trees called 'phylogenetic trees' based on similarities in the genetic sequences of different variants of the virus. Looking at these trees researchers can track how a variant spreads geographically, and also attempt to identify new worrying variants that might lead to a new wave of infections. The scale of the COVID-19 pandemic together with the global effort by clinicians and researchers to sequence SARS-CoV-2 genetic material means a library of over 13 million SARS-CoV-2 genomes now exists, making it the largest such collection for any organism.

Although phylogenetic trees of viruses have been studied for a long time, exploring the SARS-CoV-2 library presents technical and practical challenges due to its sheer size. Sanderson has developed an open-source web tool called Taxonium that allows users to explore phylogenetic trees with millions of sequences. With help from collaborators at the University of California, Santa Cruz, Sanderson built a website called Cov2Tree, that uses the Taxonium platform to allow immediate access to an expansive tree of all publicly available SARS-CoV-2 sequences. Cov2Tree enables users to visualise all SARS-CoV-2 genomes in a birds-eye view akin to a 'Google Earth for virus sequences' where anyone can zoom in on a related family of viruses down to the level of individual sequences. This can be used to compare variants and follow geographic spread.

Using Taxonium, scientists can explore how virus sequences are related to each other. They can also see the individual mutations that have occurred at each branch of the tree, and can search for sequences based on mutation, geographical location, or other factors. Interestingly, a trend appearing in the SARS-CoV-2 phylogenetic tree is the emergence of identical mutations at different branches of the tree without a common origin. These mutations may be a result of convergent evolution, a phenomenon that occurs when a mutation appears independently in different variants as it confers an advantage to the virus making such mutations more likely to persist. This means that scientists may be able to expect certain mutations to appear in more distantly related variants if they have appeared independently in several different variants already.

Overall, Taxonium is an important tool for monitoring SARS-CoV-2 genomes, but it also has broader applications. The tool can be used to browse phylogenetic trees of other viruses and organisms. Furthermore, the Taxonium website offers a way to browse a tree of life, with images and links to Wikipedia. The SARS-CoV-2 library might be the largest now, but in the future even bigger datasets will likely be available, highlighting the importance of tools like Taxonium.

bottlenecks preventing full-scale analyses. There are two broad responses possible to this issue. One is to downsample the sequences analysed in order to create a smaller dataset with which existing tools are able to work efficiently. This has been an important approach, and has allowed Nextstrain (**Hadfield et al., 2018**) to provide one of the most widely-used and important tools for exploring SARS-CoV-2 genetic diversity. Briefly, the Nextstrain pipeline downsamples sequences in a rational way, and then uses iqtree (**Minh et al., 2020**) to assemble them into a tree, assigns chronology and ancestral states to this tree using TreeTime (**Sagulenko et al., 2018**), and displays the results in a user-friendly interactive interface using Auspice (**Hadfield et al., 2018**). All three of these post-downsampling stages are bottlenecks which prevent the use of full datasets, and so Nextstrain analyses are typically limited to ~4000 sequences – sampled in a structured way to ensure they either provide an overview of the pandemic or hone in on particular sequences of interest.

The new scale of data available provides an impetus to develop new tools that are able to operate on these large datasets directly without downsampling. Recently the development of UShER (**Turakhia et al., 2021**) has permitted larger trees to be built than ever previously. UShER takes a starting tree,

built with iqtree or a similar approach, and incrementally adds sequences by maximum parsimony. For densely sampled sequencing efforts, as in the SARS-CoV-2 pandemic, such an approach still yields tree topologies of very high quality (*Thornlow et al., 2022*). To turn this distance tree into a time tree, by estimating a time associated with each node in the tree, we recently developed Chronumental (*Sanderson, 2021*) which uses stochastic gradient descent to efficiently construct chronologies from very large trees, which was not possible with previous approaches. A final necessary component is a tool for exploring these large trees, ideally in a browser.

Here, we describe Taxonium, a web-based tool for for analysing and exploring large trees. Taxonium scales to trees with millions of nodes, and allows for rapid panning and zooming using WebGL. In addition to reading Newick format trees, Taxonium can also display UShER mutation-annotated trees which capture genotype information in mutations at internal branches. It permits searching for nodes by metadata or genotype, and a range of annotation options. Taxonium is available in a server-backed mode, which in a matter of seconds loads to allow exploration of all publicly available SARS-CoV-2 sequences, and also a fully client-side mode suitable for exploring custom datasets, including those with sensitive data.

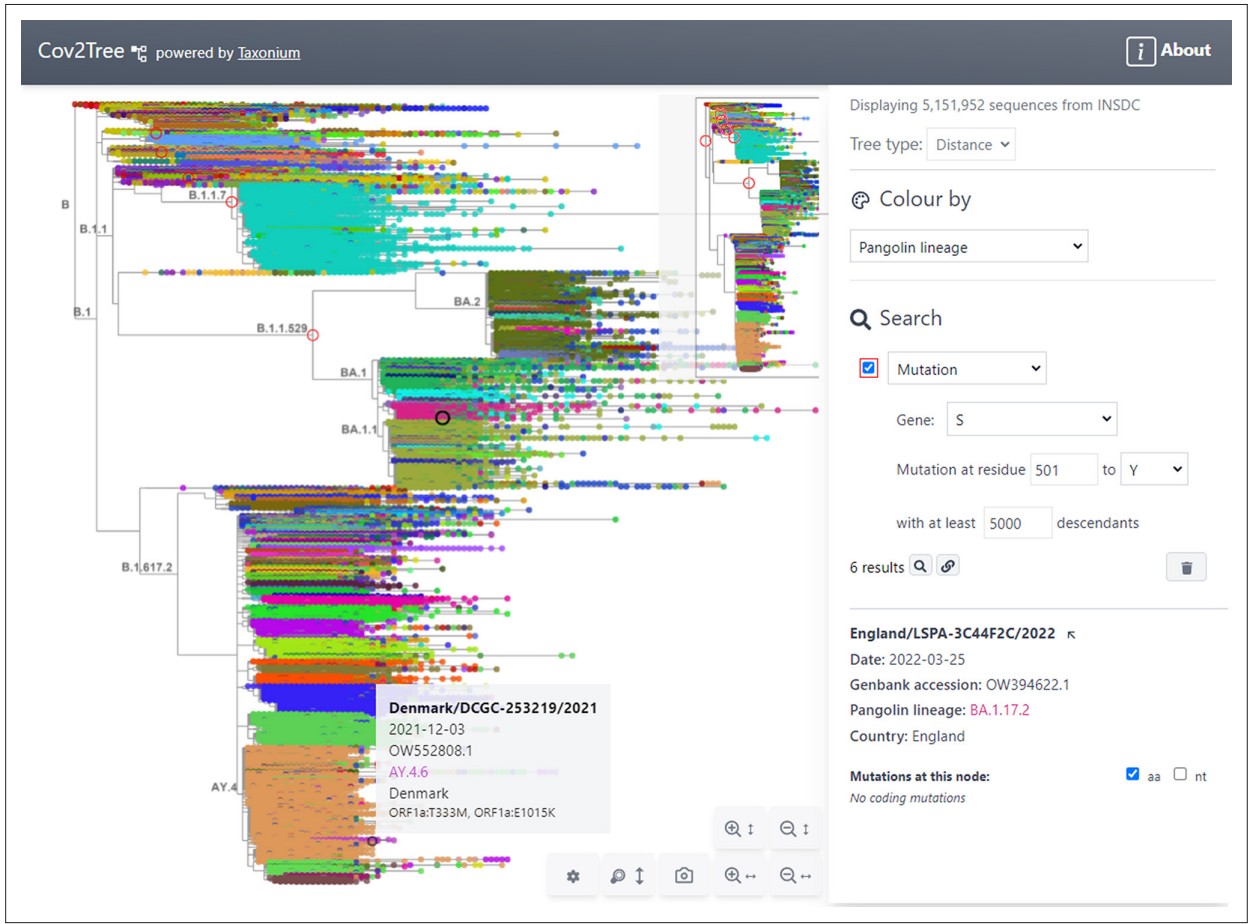

**Figure 1.** Tree exploration at unprecendented scale. This figure presents a screenshot of the Taxonium web client displaying a 5,151,952 sequence tree of SARS-CoV-2 sequences. The left hand panel shows the zoomable tree with a minimap for orientation, and the right hand panel provides options for searching for nodes and for changing the colour scheme. Hovering over a node shows information in a tooltip, while clicking on a node displays further information in the right-hand panel. A search has been carried out for mutations at S:501 to Y, filtered to such nodes with at least 5,000 descendants, which are circled in red. Genomes are coloured by their PANGO lineage.

## Results

## The Taxonium web client allows exploration of million-node phylogenies in the browser

To allow interactive examination of very large phylogenies, we built the Taxonium web client, a React application available at taxonium.org. One major bottleneck for previous approaches was the use of web technologies involving SVG or Canvas to visualize the tree, which have performance limitations. We instead use WebGL, a web standard that allows a computer's GPU to display web graphics, for efficient visualisation of the tree – we implement this using DeckGL (*Uber, 2016*). Even so, rendering every node in the tree would still be too slow, and would involve hundreds of nodes overplotted on each pixel when a tree was zoomed out. We address this by rendering a sparsified version of the tree, with the sparsification dependent on the zoom level such that only nodes that overlap other nodes are excluded (see Materials and methods). This approach allows for fast and responsive tree exploration.

The input to Taxonium is a tree (e.g. in Newick format) and, optionally, metadata. Metadata is associated with each node of the tree, and the tree can be coloured by any chosen metadata item.

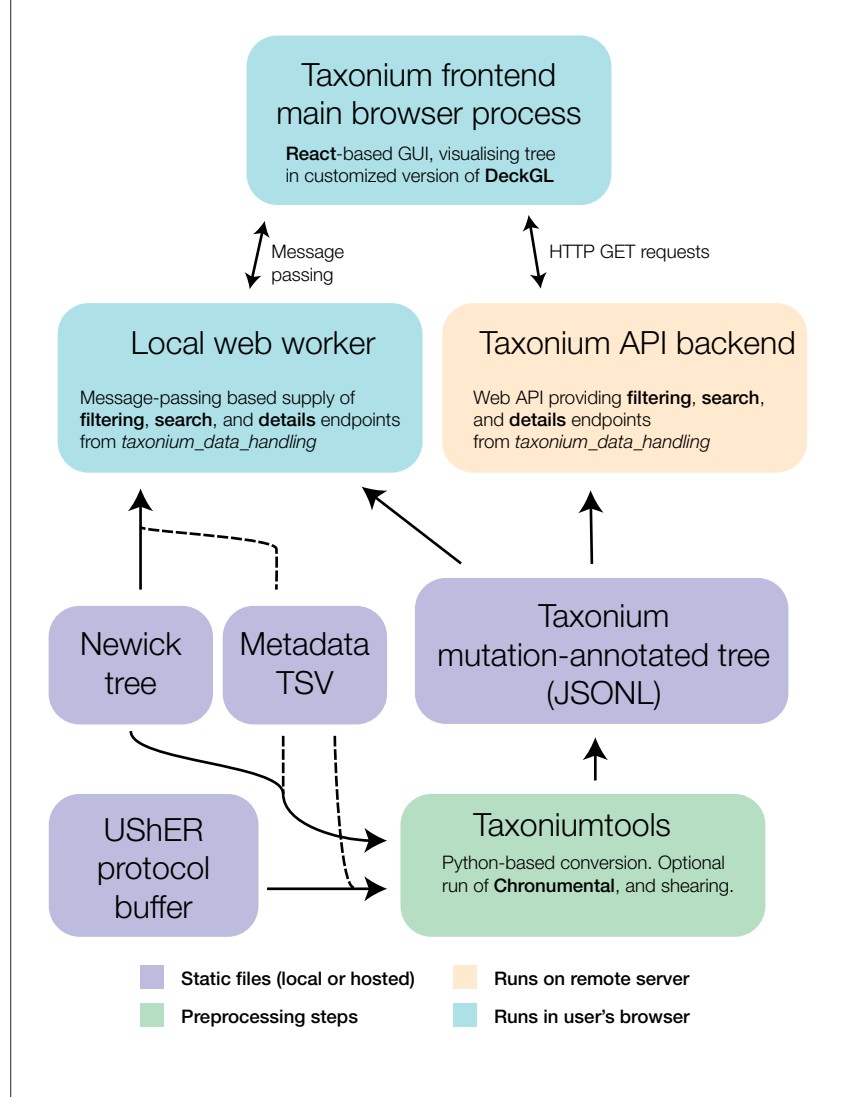

**Figure 2.** Components of the Taxonium project. At the core of Taxonium is a graphical interface that runs in the web browser. It can connect either to a web-worker running in the same browser (for purely local use) or to an API web-server that serves parts of the tree on demand. Trees can be loaded from Newick format files with metadata TSVs (tab-separated value files), or from a specialised Taxonium JSONL (JSON-lines) format which is able to capture mutation-annotated trees. Taxonium JSONLs can be produced using the Taxoniumtools library.

Colours are by default selected using an algorithm that hashes the metadata's string value into a unique colour, ensuring consistency over time without the need for comprehensive look-up tables. In addition, the tree can be searched based on the metadata, with identified nodes circled. Complex hierarchical boolean combination queries using AND, OR, or NOT are supported.

The user interface of the Taxonium web client is shown in *Figure 1*. The tree, at the left hand side of the screen, can be panned, and can be zoomed in the vertical and horizontal axes independently. The latter is a crucial feature for large trees, which are invariably much larger in their y-axis. A toggleable minimap is provided for orientation. The right hand panel allows users to search for nodes of interest, to select how the tree is coloured, and to select between a chronological and a distance tree. It also displays information about the selected node; similar information is available upon hovering over a node of interest.

In addition to accepting traditional trees, Taxonium also supports mutation annotated trees (MATs, *McBroome et al., 2021a*), generated by UShER, where internal nodes are annotated with mutations inferred to have occurred at that branch of the phylogeny (*Figure 2*). Since an MAT essentially captures the full sequence of each sample in the tree, its use as an input permits the user to colour the tree by genotype at any desired site (*Figure 3*), or alternatively to search for particular mutations in internal modes (*Figure 4*), and to filter by how many terminal nodes these mutations gave rise to.

Taxonium can allow visualisation of trees entirely in the client (*Figure 2*, l.h.s.), which is especially important for trees which may contain sensitive patient-level data. Trees are loaded either from a Newick file and TSV metadata file, or from a custom preprocessed Taxonium JSONL format combining the two. The Taxonium JSONL format contains a pre-computed layout for the tree, reducing the amount of computation required for its display, but Newick files can also be laid out in the client, which is achieved using an approach heavily based on JStree (*Li, 2021*). More expensive computational operations, such as the sparsification of the tree for display, are performed in a web worker in order to maintain a responsive interface.

## An optional server-based implementation empowers rapid analysis

The client-side only version of Taxonium is highly responsive, and permits loading trees with millions of nodes on typical consumer computers. However, the process of deserialising tree data from disk into Javascript objects in memory is a bottleneck, requiring one minute and 20 s for a tree of 5.4 million sequences. Large trees also demand increasing amounts of RAM and download bandwidth which might rule out the use of lower spec devices. To allow near instantaneous access to trees with millions of nodes on almost any internet-connected device, we built a server-backed mode for Taxonium, in which a server reads trees from disk in advance and then serves required parts of them to each client on demand (*Figure 2*, r.h.s.). The server-backed mode is more efficient, as it does not require all data to be sent to the client, and the computationally expensive operations can be performed on a more powerful machine. Using this mode, a 5.4 million sequence tree can initially load in seconds. We continue to offer the client-side mode, which has the advantage that it can be used with custom data, and especially for data that may be sensitive and not suitable for uploading.

The Taxonium backend is implemented in NodeJS using Express. It runs from the same codebase that runs in the web worker in the browser. We made substantial efforts to make this backend code as efficient as possible. Nodes are stored sorted by their y coordinates, meaning that two binary searches can be used to identify the slice of nodes that lie within a supplied window.

## The Taxonium desktop application enables local loading of particularly large trees

Like most other tools for tree exploration, Taxonium stores data on nodes in random-access memory (RAM). Most computers now have 8 GB or more of RAM, which is sufficient to store data from a very large tree. However memory limits in the browser can be significantly lower than the total system memory, and efficiency of storage can depend on the specific web browser, which can limit the size of trees which can be loaded locally in the Taxonium web client to ~6 M terminal nodes on some systems. This could be avoided by manually starting an instance of the Taxonium backend, but this would be a difficult procedure for non-expert users. In response to this issue, we built the Taxonium Desktop App.

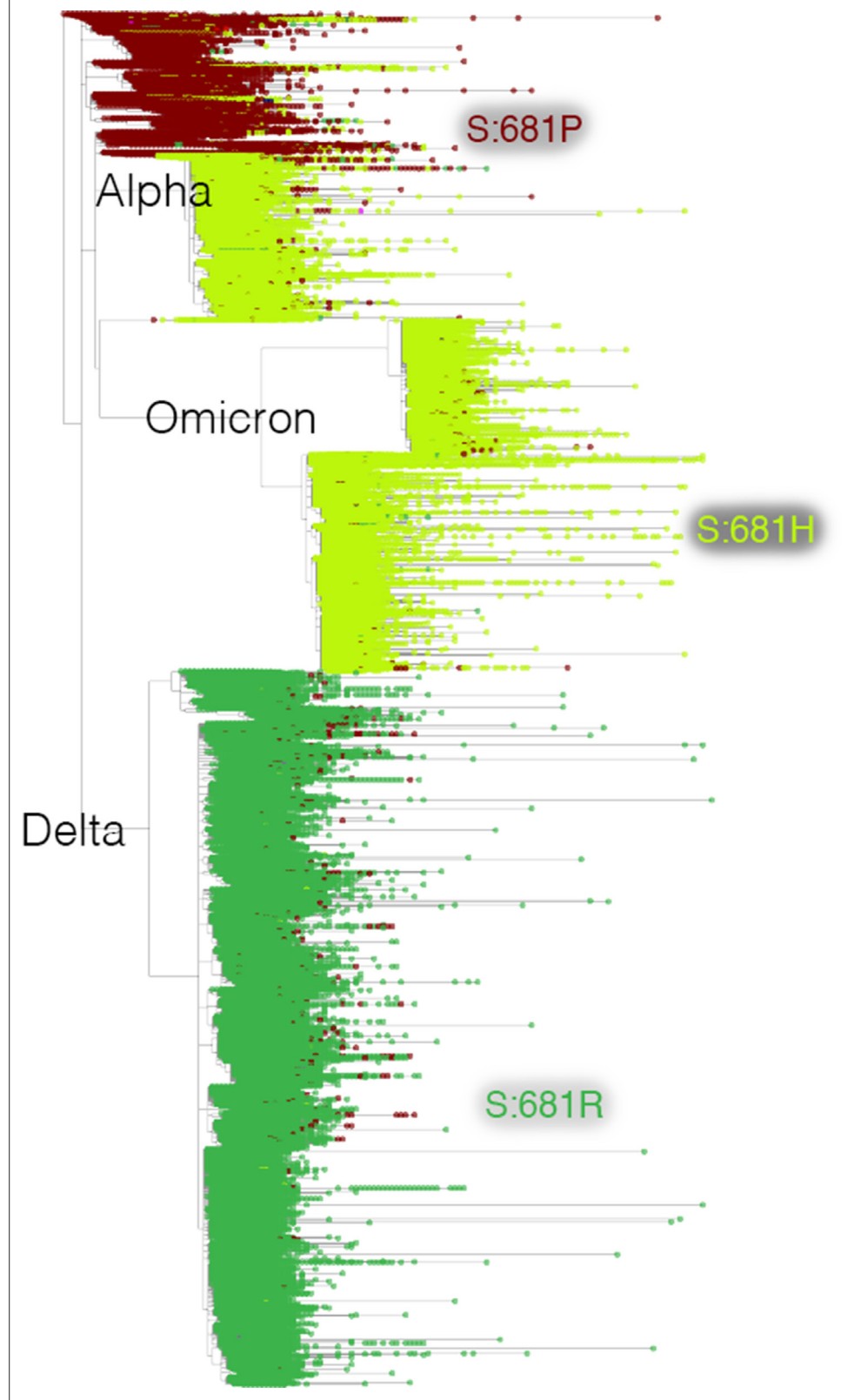

**Figure 3.** Mutation-annotated trees capture full sequence variation. In this screenshot, genotype data at position Spike 681 is overlaid onto the full SARS-CoV-2 tree with Taxonium. This analysis can be reproduced by setting the 'Color by' field to *genotype*.

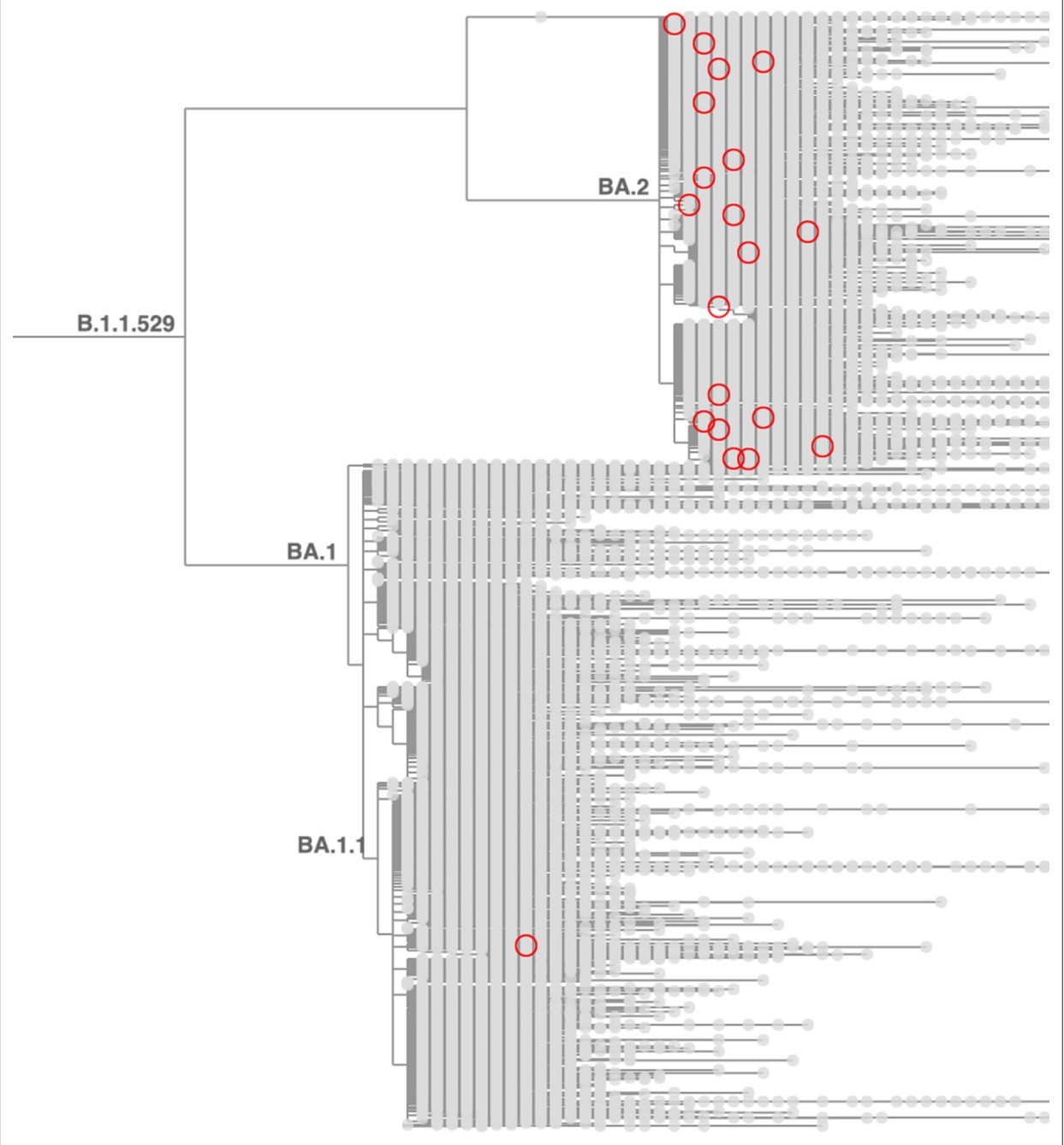

**Figure 4.** Taxonium reveals recurrent mutations in SARS-CoV-2. This figure shows a tree zoomed in on Omicron (B.1.1.529), with 1,422,024 sequences until late May 2022. Circled nodes feature a mutation at S:452 to either M or Q which have more than 10 descendants. These (independent) mutation are much more common on the BA.2 genetic background, revealing epistasis.

The app is developed using Electron and is available in binary format for MacOS, Linux, and Windows. It automatically starts an instance of the backend using a packaged version of NodeJS, which can use as much system memory as is available, and then starts a client to connect to this backend, running in Chromium. We were able to use the Taxonium Desktop App to explore a mutation-annotated phylogeny of 12.5 M sequences and metadata on a laptop with 8 GB of RAM.

## User-friendly tooling for tree generation

We provide a simple Python package, taxoniumtools, which allows straightforward generation of a Taxonium JSONL file from an UShER MAT or a Newick tree using the command-line. The Taxonium

JSONL format combines tree topology, node metadata, and mutations, in a row-wise format. The tree's structure is encoded only in the parent_id property of each node.

Taxoniumtools uses TreeSwift (*Moshiri, 2020*) for rapid loading of the Newick string located within the UShER MAT file. Optionally it can run Chronumental (*Sanderson, 2021*) as part of the building process and integrate the resulting time tree into the final JSONL file.

Full documentation for Taxonium and Taxoniumtools is available at docs.taxonium.org.

## A resource for the phylogenetic analysis of all public SARS-CoV-2 genomes

We used Taxonium to build the Cov2Tree web application (cov2tree.org), which allows users to explore the full global diversity of public SARS-CoV-2 sequences (*Figure 3*).

Cov2Tree is based on the daily-updated UShER tree built by *McBroome et al., 2021b*, which currently contains 5.6 million sequences. We provide a time tree inferred by Chronumental, and also provide a daily-updated file containing the date placements inferred by Chronumental which can allow the identification of sequences with metadata errors (*Sanderson, 2021*).

We run a backend server that supports the Cov2Tree application, meaning that the user needs only to load the data for the region of the tree on which they have zoomed in. This helps to enable analysis in lower bandwidth settings. Users can colour the tree by PANGO lineage, by sample country, or by genotype, and use searches to conduct complex queries.

The application of Taxonium to global SARS-CoV-2 datasets provides important insights into the evolution of the virus. Taxonium readily displays the number of independent times a given mutation has occurred during viral evolution, and the lineages in which these mutations evolved. This can be a key analytical approach to understand evolutionary processes such as epistasis. For example, Taxonium reveals a very heavy enrichment for occurrences of S:452 M/S:452Q are within the BA.2 lineage (*Figure 3*), providing evidence of epistatic interactions between this position and BA.2 defining mutations. Similar effects are seen at S:212.(see this tree).

Taxonium has also been used to probe regional introduction events and transmission clusters during the SARS-CoV-2 pandemic (*McBroome et al., 2022*), to identify a convergent origin for two sets of mutations which each create a new ORF (*Mears et al., 2022*), and to examine new tools for pandemic-scale tree reconstruction (*De Maio et al., 2022*).

## Taxonium has applications beyond SARS-CoV-2

We have used Taxonium to create a tool that allows exploration of the NCBI Taxonomy database (*Federhen, 2012*). The resulting visualisation, which can be found at taxonomy.taxonium.org, allows interactive viewing of the taxonomic relationships between 2.2 million species, as well as searches (*Figure 5*).

We have also collaborated with the Serratus project (*Edgar et al., 2022*) to allow visualisation of trees of viral RdRP sequences identified in a search of petabases of reads from the Sequence Read Archive. Each viral order and family at serratus.io/trees now provides an option to open a custom Taxonium tree.

With the arrival of the 2022 monkeypox outbreak in Europe, we launched mpx.taxonium.org (*Figure 5*) to allow exploration of an open genomic dataset from LAPIS (*Chen, 2022*).

## Taxonium scales to larger trees than any existing tool

We argue that Taxonium is currently the tool that best scales to the largest trees. To examine this we here compare a number of tools for their ability to load a very large tree. We stress that many of these tools were likely not specifically designed to be able to load trees of this size, unlike Taxonium.

We used the example of a Newick tree derived from a recent UShER public tree (*McBroome et al., 2021b*) featuring 5,326,538 sequences(tree can be downloaded here). On a Macbook Pro (2018 version, 2.3 GHz quad-core i5), Taxonium loaded this tree from the Newick file in 31 s.

Archaeopteryx (*Han and Zmasek, 2009*) took >5 min to load this tree with stack size set to 5 GB, and then was essentially unresponsive. Empress 1.20 (*Cantrell, 2021*) did not load this tree in the browser in any reasonable length of time. We were unable to upload this file to iTOL (*Letunic and Bork, 2021*) or the ETE Tree Explorer (*Huerta-Cepas et al., 2010*).

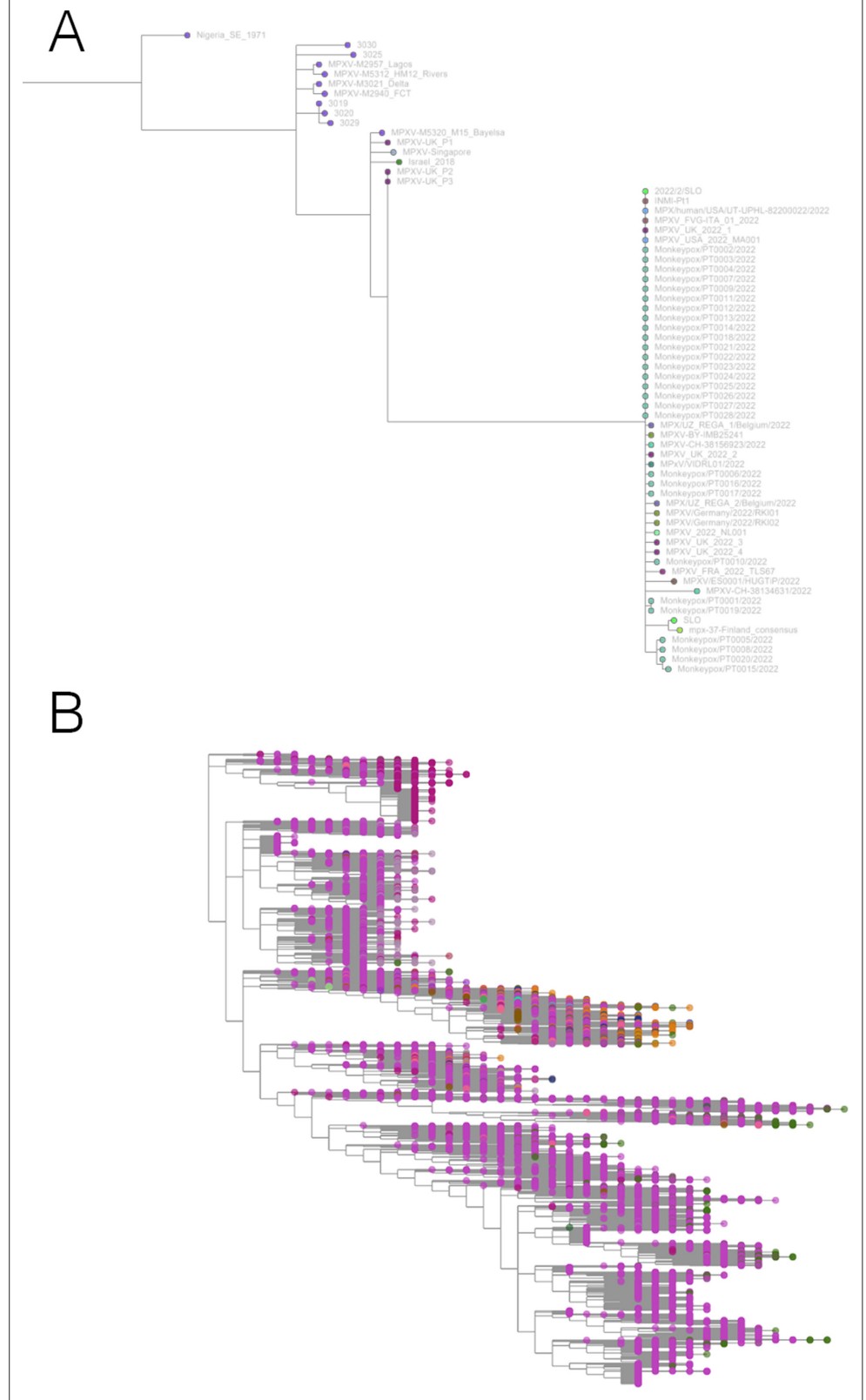

**Figure 5.** Taxonium has applications beyond SARS-CoV-2. (**A**) Phylogeny of monkeypox sequences at mpx. taxonium.org. The large clade represents the current outbreak (**B**) NCBI's taxonomy of 2.2 M species as explorable at taxonomy.taxonium.org.

We attempted to load the same file on Microreact.org (*Argimón et al., 2016*) (which now uses Phylocanvas.gl), but the screen locked up and had not loaded within a reasonable time. A Phylo-canvas.GL demo does show that the library (*Abudahab et al., 2021*) is able to load a SARS-CoV-2 tree with >1 M leaves, panning relatively slowly on our hardware.

We found it difficult to evaluate the performance of Dendroscope (*Huson et al., 2007*): we were able to open this file – for which we increased its allotted RAM to 5 GB. The tree loaded in 1 min and 23 s. However, it rendered as a single black rectangle which only began to resolve upon zooming in substantially, making it difficult to extract insights about the tree. Dendroscope used all available RAM whereas Taxonium used 1.7 GB. In general, Taxonium felt substantially more responsive.

It must be noted that all these tools have different feature sets. Many of the tools above allow for features absent from Taxonium, such as re-rooting, displaying trees radially, collapsing subtrees, and more. Support for these features to an extent trades-off against scale. Conversely, Taxonium is unique among these tools in supporting mutation-annotated trees, and perhaps in its ability to toggle to a time tree – features shared with, and inspired by, Auspice (*Hadfield et al., 2018*) – which is limited to phylogenies of <~10,000 tips (around 0.2% of Taxonium scale). In sum, we argue that Taxonium is the only tool able to fluidly explore the largest trees in current use on consumer-grade hardware.

## Discussion

The increased scale of recent genomic datasets has prevented existing tools from being used to explore entire phylogenies in the case of SARS-CoV-2. While unprecedented, these SARS-CoV-2 data-sets likely provide a preview of future data volumes for many other organisms given the decreasing costs of sequencing and increasing infrastructure for genomic surveillance.

Taxonium allows exploration of these large phylogenies, and is also one of few tools to allow the visualisation of a mutation-annotated tree capturing all known sequence diversity in an organism. It is open-source and highly configurable for use with any tree. We have used Taxonium to build the Cov2Tree website, which has been widely used to explore the global diversity of SARS-CoV-2.

Some tools for phylogenetic trees are designed primarily for generating static figures suited for publications. These may offer features Taxonium lacks, such as the ability to visualise phylogenies in a circular layout, and the ability to output vector graphics. In contrast, Taxonium is intended primarily for exploring a tree dynamically – in the case of very large trees, this is often the only way to extract useful information.

One challenge in this new era of vast datasets is the variable quality of sequences, and therefore resultant phylogenies. Artifacts in sequences (e.g. *Heguy et al., 2022*; *Sanderson and Barrett, 2021*; *Sanderson et al., 2021*) are often systematic, and in such cases can create entire clades brought together by shared errors. Existing efforts to tackle these issues have been important, for example pooling knowledge of known sites of common spurious mutations (*De Maio et al., 2020*). The development of new pipelines and assemblers that minimize artifacts will be important in the future, and the development of tree-building algorithms that are able to consider the possibility of variation due to artifacts, or pre-tree masking steps that mask different sites in each sequence, may be valuable. Widespread deposition of raw reads would allow systematic assembly by pipelines designed to mini-mise errors. In Taxonium, we provide a search option that highlights 'revertant' mutations in which a branch apparently mutates back towards the reference genome. In large SARS-CoV-2 phylogenies, many of these mutations appear to in fact represent sequencing artifacts, or reversion from them, and so this approach may help identify areas of a tree that have technical issues.

Even where sequences are accurate, phylogenetic topology is often uncertain, and finding ways to communicate this at scale, building on prior work (*Bouckaert, 2010*) would be useful. A further chal-lenge is the vastly different densities of sampling in different geographic regions. Because Cov2Tree does not downsample sequences from countries which are able to sequence a greater proportion of their cases, the number of tips in a particular region of the tree is not always indicative of the size of an outbreak, and in some cases even inferences of the directionality of migration may be confounded. There would be value in the development of techniques that allow visual normalisation of trees for sampling biases, which might allow for less biased phylogenetic representations without downsampling.

Taxonium is an ongoing project to create and maintain a tree viewer able to scale to the latest phylogenies. It forms part of an ecosystem of open-source tools that together turn an avalanche of

sequencing data into actionable insights into ongoing evolution. We welcome contributions from the community and hope that the features described here will be useful to those working on epidemiological surveillance and outbreak response.

## Methods

### Data sources

In order to be able to share trees through a web application, we present data at Cov2Tree and in this paper derived from SARS-CoV-2 sequences made available openly in the INSDC databases. We are fortunate to be able to use a pre-computed UShER tree also made available openly on a daily basis (*McBroome et al., 2021b*).

### The Taxonium web client

The Taxonium web client is implemented in React. All source code is available at https://github.com/theosanderson/taxonium/tree/master/taxonium_web_client (*Sanderson, 2022*). The core of the visualisation runs in DeckGL, which is heavily modified to allow independent zooming in X and Y. The data consumed by DeckGL is generated either in a web worker (local mode) or in from an external API running the Taxonium backend (remote mode).

In either case, the key endpoint provides a sparsified tree, a pruned version focused on nodes in the currently visible region, with overlying nodes reduced (see *Reducing overplotting* below). Outputting a full pruned tree means that the genotypes of all terminal nodes can be computed by traversing the mutations associated with each node.

The returned dataset includes all metadata for each included node, meaning that users can select any item to colour nodes by, or alternatively can colour nodes by genotype.

Alternatively the *Search* endpoint allows users to search across the whole dataset, with nodes matching the search circled in the interface.

Taxonium is capable of storing two different distance metrics in the same tree, allowing users to switch between the genetic distance and the temporal phylogeny.

### Taxonium backend

The Taxonium backend is implemented in NodeJS in an Express-based framework. Full source code is available at https://github.com/theosanderson/taxonium/tree/master/taxonium_backend. The core code for filtering and searching that runs in the web-worker for the front-end is reused through a shared package, *taxonium_data_handling*.

The backend keeps the full tree in memory, then serves filtered slices of it to users on demand. In the case of the SARS-CoV-2 dataset, the server backend requires around 5 GB of RAM per server instance. We run Cov2Tree on an autoscaling Kubernetes cluster.

### Reducing overplotting

#### Nodes

To minimise both the overhead for displaying data in DeckGL, and the amount of data that needs to be transferred from a backend to the client, it is necessary to filter the tree so that we do not aim to draw every single node. Drawing every single node is unnecessary given that a tree with 5 M terminal nodes contains more nodes than there are pixels available on a typical screen, meaning that many nodes lie on top of one another. Therefore, we aim to filter out some of these overlying nodes while ensuring that the user sees essentially the same tree that they would have done without filtering.

This approach is assisted by the fact that the tree's layout is pre-calculated, with each node assigned an X and Y value which is stored in the Taxonium JSONL (or generated at load-time from a Newick file).

When the user zooms to a particular region of the tree, we create a bounding box, which is a little larger than the area visible on the screen. We then calculate a 'precision' value for the X and Y axes, which is a value <1, inversely proportional of the dimensions of the bounding box in X and Y, so that as the bounding box targets a smaller area the precision value increases. We initially select all the nodes which are within the bounding box, and then filter down overlying nodes, using the following approach which is designed to be as streamlined as possible to minimise computational work:

- Create a variable to store already-included positions (which are indexed as `included_points[rounded_x][rounded_y]`)
- Iterate over nodes
  - For each node calculate a 'rounded position', with an x-value of `Math.round(node.x * precisionX) / precisionX` and the corresponding y-value.
  - If this rounded position is *not* among the already-included positions then:
    - include this position in the output
    - add this position to the already-included positions

## Search

In order to ensure that search results are always comprehensive, but at the same time to avoid over-plotting, we take the following approach:

- Searches are performed across every single node on the tree to identify a set of nodes that match the search. The total number of matches is displayed in the client.
- If fewer than 10,000 matches are detected, these are simply directly displayed in the client as circles
- If more than 10,000 matches are detected, the results are first sparsified using the same method described above for nodes, and then displayed.
- Upon zooming or panning, the sparsification is repeated for the new bounding box.

## Taxoniumtools and tree processing

The *taxoniumtools* package is written in Python and distributed with PyPI. Tree manipulation is performed with TreeSwift (**Moshiri, 2020**). The package includes the ability to number internal nodes, with a numbering scheme that matches UShER newick exports. An optional feature, used for display on Cov2Tree, 'shears' off outlier nodes with very few descendants, which often represent sequencing errors or occasionally recombinants. This pruning can help to make large trees more interpretable.

## Data availability statement

The input mutation-annotated trees from publicly available SARS-CoV-2 genomes are described in **McBroome et al., 2021b** and made available at http://hgdownload.soe.ucsc.edu/goldenPath/wuhCor1/UShER_SARS-CoV-2/.

The code for Taxonium is available at https://github.com/theosanderson/taxonium (**Sanderson, 2022**).

## Acknowledgements

Phylogenetic analysis of viral genomes is only possible thanks to a community of researchers and clinicians who take viral samples, generate genomes, and submit to sequence databases such as the INSDC databases and GISAID, and I am very grateful to all contributors. I am indebted to Richard Goater for development advice and to Angie Hinrichs for maintenance of the underlying tree that powers Cov2Tree. This work was supported by the Wellcome Trust (210918/Z/18/Z) and the Francis Crick Institute which receives its core funding from Cancer Research UK (FC001043), the UK Medical Research Council (FC001043), and the Wellcome Trust (FC001043). This research was funded in whole, or in part, by the Wellcome Trust [210918/Z/18/Z, FC001043]. For the purpose of Open Access, the author has applied a CC BY public copyright licence to any Author Accepted Manuscript version arising from this submission.

## Additional information

### Funding

| Funder | Grant reference number | Author |
| --- | --- | --- |
| Wellcome Trust | 210918/Z/18/Z | Theo Sanderson |
| Wellcome Trust | FC001043 | Theo Sanderson |

| Funder | Grant reference number | Author |
|---|---|---|
| Cancer Research UK | FC001043 | Theo Sanderson |
| Medical Research Council | FC001043 | Theo Sanderson |

The funders had no role in study design, data collection and interpretation, or the decision to submit the work for publication. For the purpose of Open Access, the authors have applied a CC BY public copyright license to any Author Accepted Manuscript version arising from this submission.

## Author contributions

Theo Sanderson, Conceptualization, Software, Formal analysis, Funding acquisition, Validation, Investigation, Visualization, Methodology, Writing – original draft, Project administration, Writing – review and editing

## Author ORCIDs

Theo Sanderson http://orcid.org/0000-0003-4177-2851

## Decision letter and Author response

Decision letter https://doi.org/10.7554/eLife.82392.sa1
Author response https://doi.org/10.7554/eLife.82392.sa2

## Additional files

### Supplementary files

- MDAR checklist

### Data availability

All code is available on GitHub. Data was not generated as part of this study. Data sources are indicated in the manuscript and raw data is available in all cases, without the need for requests.

The following previously published dataset was used:

| Author(s) | Year | Dataset title | Dataset URL | Database and Identifier |
|---|---|---|---|---|
| McBroome J, Bryan T, Hinrichs AS, Kramer A, De Maio N, Goldman N, Haussler D, Corbett-Detig R, Turakhia Y | 2021 | A Daily-Updated Database and Tools for Comprehensive SARS-CoV-2 Mutation-Annotated Trees | http://hgdownload.soe.ucsc.edu/goldenPath/wuhCor1/UShER_SARS-CoV-2/ | UCSC, UShER_SARS-CoV-2 |

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
