## [Editor Report]

Sanderson developed novel interactive software for visualizing phylogenetic trees representing millions of sequences. This is a fundamental advance over previous software that is typically limited to trees with a few thousand tips. Taxonium has been used intensively by the virus evolution community over the past months and has thus already proven its utility and performance.

---

## [Decision Letter]

**Decision letter after peer review:**

Thank you for submitting your article "Exploration of million-sequence viral phylogenies with Taxonium" for consideration by *eLife*. Your article has been reviewed by 3 peer reviewers, including Richard A Neher as Reviewing Editor and Reviewer #1, and the evaluation has been overseen by a Reviewing Editor and Neil Ferguson as the Senior Editor. The following individuals involved in the review of your submission have agreed to reveal their identity: Art FY Poon (Reviewer #2); James Hadfield (Reviewer #3).

Essential revisions:

All reviewers agreed that Taxonium enables the interactive exploration of phylogenies orders of magnitude larger than what was possible before and represents an urgently needed technological advance given the large volume of SARS-CoV-2 data that was generated over the last 3 years. The reviewer comments mostly represent clarifications and suggestions for a more precise description of the tool, its scientific context, or prior work. The most essential revision is the following:

We would like to see a more detailed description of the "sparsification" algorithm. How are the tips selected that is being rendered? how does this interact with the search? Could this (possibly as in future development) be made dependent on metadata (hosts, geography)?

In addition to improvements to the manuscript, the reviewers have identified a number of points that might be useful to consider during the future development of Taxonium.

*Reviewer #3 (Recommendations for the authors):*

Terminology: the term "nodes" is used in the manuscript to represent tips (as far as I can tell), whereas this typically refers to both internal nodes and terminal nodes (tips).

Examples: "Taxonium scales to trees with millions of nodes' and "NextStrain analyses are typically limited to ~4,000 nodes". I suggest changing to "tips" or "terminal nodes".

The introduction talks about 11 million sequences, but Cov2Tree only uses the NCBI set (~6 million). I presume this is due to GISAID data-sharing conditions and not a technical limitation of Taxonium. This is a delicate situation however it was strange to introduce the former but then present the latter.

The following points should be thought of as suggestions to improve Taxonium rather than requirements for publication:

– A legend to explain the colours is needed – you essentially add this in Figure 3 but it should be part of the software.

– Panning of trees (x + y direction) is intuitive but it's easy to get lost and pan the tree off the screen! It should be possible to prevent panning outside the bounds of the tree.

---

## [Author Response]

Reviewer #3 (Recommendations for the authors):Terminology: the term "nodes" is used in the manuscript to represent tips (as far as I can tell), whereas this typically refers to both internal nodes and terminal nodes (tips).Examples: "Taxonium scales to trees with millions of nodes' and "NextStrain analyses are typically limited to ~4,000 nodes". I suggest changing to "tips" or "terminal nodes".

Thank you for pointing this out. I have changed the Nextstrain sentence to refer to sequences, and am sorry for that error. (I have also belatedly corrected my mis-capitalisation of Nextstrain in the final manuscript file!) I then went through the manuscript and considered each use of the word "node", making some changes. I suspect I have kept more of the bare term "node" than you might prefer. The way in which Taxonium is implemented means that terminal and non-terminal nodes behave in almost exactly the same way. Although in Cov2Tree they are not, internal nodes can be associated with metadata, and can (using the settings cog) be coloured by that metadata. Limitations on tree size relate more with the total number of nodes than the number of tips. That is the rationale for describing things in those terms, but I do also see that readers have a better intuition for tip-number. In general for the polytomous SARS-CoV-2 trees that we consider here the numbers are surprisingly similar.

The introduction talks about 11 million sequences, but Cov2Tree only uses the NCBI set (~6 million). I presume this is due to GISAID data-sharing conditions and not a technical limitation of Taxonium. This is a delicate situation however it was strange to introduce the former but then present the latter.

This is a fair point. Taxonium has been designed to scale to the full size of datasets available globally (i.e. including GISAID sequences as well as those from INSDC). A number of groups are privately using Taxonium in that way. In the updated manuscript I have added a paragraph on the newly developed Electron-implementation of Taxonium and there I discuss loading a tree with 12M nodes, which may help with this point.

The following points should be thought of as suggestions to improve Taxonium rather than requirements for publication:– A legend to explain the colours is needed – you essentially add this in Figure 3 but it should be part of the software.

Thank you for flagging this. (As you'll be aware,) this isn't completely trivial because e.g. there are more than 2,000 PANGO lineages and in theory almost all could be semi-visible on the initial load of the SARS-CoV-2 tree. But it's also not impossible: one would count the number of each colour, and sort from the most frequent, truncating at some point. I will track this in https://github.com/theosanderson/taxonium/issues/438.

– Panning of trees (x + y direction) is intuitive but it's easy to get lost and pan the tree off the screen! It should be possible to prevent panning outside the bounds of the tree.

That's a great idea. I will track this in https://github.com/theosanderson/taxonium/issues/439.